# The Social Fabric of Cheese Agroindustry: Cooperation and Competition Aspects

**María Camila Rendón-Rendón** [1,*] , **Juan Felipe Núñez Espinoza** [2,*] ,
**Ramón Soriano-Robles** [3] , **Valentín Efrén Espinosa Ortiz** [4] , **Luis Manuel Chávez Pérez** [4]
**and Randy Alexis Jiménez-Jiménez** [4]

1   Doctorado en Ciencias Agropecuarias, Universidad Autónoma Metropolitana, Unidad Xochimilco,
    Calzada del Hueso 1100, C.P. 04960 Ciudad de México, México
2   Especialidad de Posgrado en Estudios del Desarrollo Rural, Colegio de Postgraduados Campus Montecillo,
    km 36.5 carretera México, Texcoco, C.P. 56230 Texcoco, México
3   Departamento de Biología de la Reproducción, Área de investigación en Reproducción Animal Asistida,
    Laboratorio de Recursos Socioambientales y Sustentabilidad, Universidad Autónoma Metropolitana,
    Unidad Iztapalapa, San Rafael Atlixco 186, C.P. 09340 Ciudad de México, México; ramon@xanum.uam.mx
4   Departamento de Economía, Administración y Desarrollo Rural, Facultad de Medicina Veterinaria y
    Zootecnia, Universidad Nacional Autónoma de México, Avenida Universidad 3000, C.P. 04510 Ciudad de
    México, México; veoee1@hotmail.com (V.E.E.O.); luischavez_80@hotmail.com (L.M.C.P.);
    alexis.j2@gmail.com (R.A.J.-J.)
*   Correspondence: mcrendon@gmail.com (M.C.R.-R.); nunezej@colpos.mx (J.F.N.E.)

**Abstract:** The aim of this study was to analyze the relational social structure of the cheese factories based on an agribusiness territory of Mexico through social network analysis (SNA) in order to understand how different types of agroindustries coexist and endure. Participant observation and semi-structured interviews were carried out in 17 cheese agribusinesses located in the area of San José de Gracia, Michoacán (Mexico), in order to get insight into the family, inter-company, commercial and technical ties they have built. The SNA showed that in the community there is a meso-system where different cheese companies that produce either natural, imitation or both cheeses converge and coexist. These agroindustries make up a complex social structure composed of 1717 actors, comprising a dispersed network with low connectivity (density <0.5%) due to the commercial nature of the relationships (95.9%). Simultaneously, an underlying network with a higher density (1.73%) was also evident, enriched by kinship and friendship ties that create cooperation and trust among the parties through 136 reciprocal tangible and intangible exchanges. Despite the differences and asymmetries of cheese agribusinesses in this community, the social structure they form behaves like a 'local neighborhood' where everyone knows everyone, and everyone coexists, competes and shares with one another, allowing them to be sustainable in the marketplace. This study provides important lessons for institutions that promote competitiveness and local development, because it shows that in order to achieve sustainability of agroindustrial companies, it is important to recognize and promote long-term social structures based on trust, friendship and reciprocity.

**Keywords:** cheese; coexistence; kindship; reciprocity; social network

## 1. Introduction

The establishment of the neoliberal model around the world brought an unsustainability to socioeconomic structures by privileging economic issues that encourage the accumulation of capital [1]. These societies are defined by (1) an extreme deregulation of economic activities; (2) complete or partial state withdrawal from fulfilling social needs and demands, leading to privatization and

individualization of society; and (3) the creation of individuals and companies with no social reference points, except for commercial reference points [2–4]. From this point of view, it would seem likely that the social contract would tend to be cancelled, leaving a scenario where both individuals and companies would devote themselves only to competing with one another in order to achieve a maximum economic benefit.

However, in some territories social conglomerates of enterprises can be observed where several factors combine with individual and collective needs and interests, and both competition and collaboration are inputs that balance the commercial component in companies. The structural makeup of the cheese industry in Mexico is one such example. It is one of the oldest manufacturing industries in the country [5] (p. 19), and generates numerous rural micro, small and medium enterprises (MSME) that continue to survive in current competitive markets. According to the National Institute of Statistic and Geography (INEGI) [6], the manufacturing sector (3.5%) is the industry with the lowest percentage of bankrupt companies in the country and the highest business life expectancy at birth (9.5 years on average). MSMEs are reticular systems developed on a social platform to bond, communicate and innovate their internal social structure. This allows them to build and deploy social abilities across a certain territory and design protection mechanisms against possible contingencies [7,8]. This results in an enterprise survival rate achieved only by a small group of companies, which have been able to build certain management reticular capacities that would be worth analyzing and replicating in less fortunate cases. In this context, the Mexican cheese industry, a generator of MSMEs, has complex and structural qualities that could teach important lessons for local development.

These survival rates of the cheese manufacturing industry are reflected in the increase in production and varieties of cheese in the country in recent years. Between 30 and 40 varieties of cheese are estimated to be produced in Mexico [9–11], and cheese consumption and production have been growing in recent years—between 2009 and 2017, cheese production grew by 79.2%. The year 2017 was marked by the highest historical production in the country, with 395,718 tons, which accounted for 74.9% of national consumption per capita [12].

This diversity of cheese varieties is also influenced by commercial openness and the globalization of global agri-food systems [13], which have had an impact on both consumption and production. In Mexico, this was first observed after structural reforms implemented after the country entered into the General Agreement on Tariffs and Trade (GATT) in 1986, and was completed in 1994 with the North American Free Trade Agreement (NAFTA). These reforms resulted in the promotion of foreign investment and the reduction of tariffs [14,15], which led to the entry of cheaper products and a greater access to other technologies that scarcely existed in the country that have promoted technological changes in agroindustries and led to a competitive agri-food sector [16]. Actually, the cheese agroindustry is characterized by an increase of imports of dairy products both for industrial use and for final consumption (mainly cheeses) [17], as well as generating an economic and technological concentration among few cheese agribusinesses [18].

These effects of the structural reform have established a heterogeneous Mexican dairy market composed of different types of companies: transnational, national and family or artisanal (many of them informal), each with different objectives, products, technologies and strategies [19]. This situation is not unique to Mexico, and the Panamerican Dairy Federation (FEPALE) [20] reports that in Latin American countries generally the dairy agroindustry is made up of a small number of large transnational and national companies together with a large number of formal and informal MSMEs. Accordingly, in Mexico, small-scale production companies (close to the subsistence level and with minimal possibilities of reproduction of capital) coexist and compete with large-scale companies (national and transnational) that use state-of-the-art techniques [21] and commercial structures based on large productive chaining and strategic alliances [17].

The coexistence between different scales of enterprises has occurred because cheese agroindustries looking for greater competitiveness in the market, and to maximize profits, have begun to produce cheaper cheeses by adding raw materials other than fluid milk to their manufacturing processes,

including milk powder, whey, caseins, starches, vegetable fat, and so on [15]. They began to produce imitation cheese, also referred to as non-natural, analogous, substitute, alternative, simulated or fake cheese, containing non-dairy ingredients that totally or partially replace milk [22].

These substitute cheeses have invaded the Mexican market and are in the greatest demand, being the most widely consumed [13,22]. While there has been an increase in cheese production and consumption in the country, these imitation products account for 80% of the market [23]. The growing sales of these products are closely linked to the evolution of convenience foods, for which the offer is expanded, and the cost reduced [24] by between 30% [25] and 43% [26]. This is compounded by the progressive interest of consumers for foods containing less total and saturated fat and cholesterol and fewer calories [24].

Consequently, imitation cheeses have displaced natural ones from large stores and cities, forcing many families or artisan businesses to shift to production of imitation products in order not to disappear [25]. These changes in the forms of production have consequences, since natural cheeses (like milk) play an important role in the Mexican rural economy, where there are still some milk-cheese production basins as well as local/regional markets that privilege consumption of products with local and ancestral know-how [13,19]. This favors a set of social and economic dynamics around cheese production and commercialization [10].

In this regard, in a scenario determined only by commercial and competitive factors, one might expect the large cheese agribusiness to have long ago displaced the small ones, or all of the latter to have been transformed to be competitive in order to remain present in the market [25,27]. However, this has not happened, and some studies in Mexico have identified that there are territories where the artisanal cheese factories coexist along with the industrial ones [28,29], although they do not provide substantive explanations indicating how they coexist.

Relying on production/competition-based approaches, some analyses highlight that to subsist and obtain comparative advantages, companies must displace the competition and other forms of production through complex modes of relationships, interdependence, hostility, leadership and power [30]. These types of approaches consider communities of cheese producers to be distrustful, disorganized, isolated and lagging in innovation, and assert that collective mercantile organization is the only way of collaboration to obtain real benefits [4,28,31].

These conventional approaches, however, do not manage to recognize the various organizational social dynamics present in every community that influence the production process. These approaches fail to recognize cheese factories as part of a meso social system, where complex and dynamic social structures converge and/or disrupt and are composed of commercial, social, technical-pedagogical, provisioning, infrastructural, synergistic and communication organizations, among others.

Against this backdrop, approaches such as network analysis could consider that cheese production and marketing in Mexico is structured in different organizational and social agglomeration levels, all of which are directly or indirectly linked. In line with this, cheese should be viewed as the result of social construction. Therefore, discussing Mexican cheese agribusiness entails referring to multiple actors and ties constituted in a social system organized around cheese production.

The different types of cheese agribusinesses based in Mexico have different goals and strategies to thrive and remain, either individually or jointly, in the market [19,21]. However, all of them build links and interdependence between key elements: production, marketing and consumption [32]. This results in the participation of many actors in the sector, namely, suppliers (supplies, raw materials, machinery, equipment), transformers, intermediaries and end consumers, generating either consensus synergies or ruptures. The comparative advantages of cheese agribusinesses derive from the deliberate creation of relationships between the various actors involved in the system [31,33]. These relationships are built through the promotion of trust, strengthened by geographical proximity [34–37] and, therefore, the construction of social networks based on cooperative–competitive strategies [38,39] between the different companies that make them up and make them sustainable [33]. As observed in another type of agribusiness and production chains, mainly in Africa, it has been found that geographical proximity

and family and/or friendship relations generate bonds of trust and cooperation between the different actors, creating a dynamic that is not only based on competition [37,40–43]. In the agribusiness of cheeses in America and particularly in Mexico, there is no evidence of these kinds of relationships or social structures. Thus, the cheese agroindustry in Mexico is possibly made up of much deeper and less visible social structure than the commercial or technical industry, which generates clusters and relationship structures and experiences that are not visible under conventional methodological approaches. Therefore, there is a socio-structural complex, immersed in a multifunctional dynamic, could be analyzed [11].

Despite this, there are few studies that have sought to visualize this type of social structure and obtain a better understanding of how the relationships between actors are maintained and made to coexist. The studies carried out about the agroindustry of cheese in Mexico have focused on identifying the actors that promote the dissemination of know-how of artisanal cheeses [28] and the factors that support the strengthening of the production of genuine cheeses on formal organizations [29,44], but they have not have identified aspects that support the social basis where competition cohabitates with cooperation and allows the coexistence of different types of companies that are possibly linked to the success of these projects.

A social structural approach could explain the factors and complexity that permit coexistence and enduring of different agroindustries in Mexico and, in particular, cheese agroindustries in the locality of San José de Gracia, Michoacán. The cheese agroindustry in this territory has centuries of tradition, and has developed manufacturing capabilities as well as substantial production and market structures [45]. Between 1943 and 1956, there were about 100 cremerías (local name for the cheese agroindustries) in the area, located both in this area itself and in its surroundings [46]. According to McDonald [47], by 1997 there were 60 agribusinesses, ranging from extremely small and rudimentary operations to large and technologically sophisticated. The cheese agroindustries have been and still remain one of the main sources of income for the municipality [26,45,48] where different types of cheese agroindustries coexist [26,45,47].

In order to contribute to an understanding of the coexistence and sustainability of the diversity of companies in the competitive cheese market, this study aims to analyze the relational social structure of the different cheese agroindustries in San José de Gracia, Michoacán, Mexico, which could teach important lessons about institutions that promote competitiveness and local development.

## 2. Materials and Methods

The study was conducted in San José de Gracia, in the municipality of Marcos Castellanos (Michoacán, Mexico) (Figure 1), located in the northeast of the state, at an altitude of 2000 m above sea level. The municipality is in the milk basin of Ciénaga de Chapala, which includes the northeastern region of Michoacán, bordering the state of Jalisco.

This research study was carried out following two methodological lines: (1) a mixed approach (quantitative and qualitative) used to profile the population of cheese producers to be studied, and (2) a topological and structural perspective, enabled by the SNA.

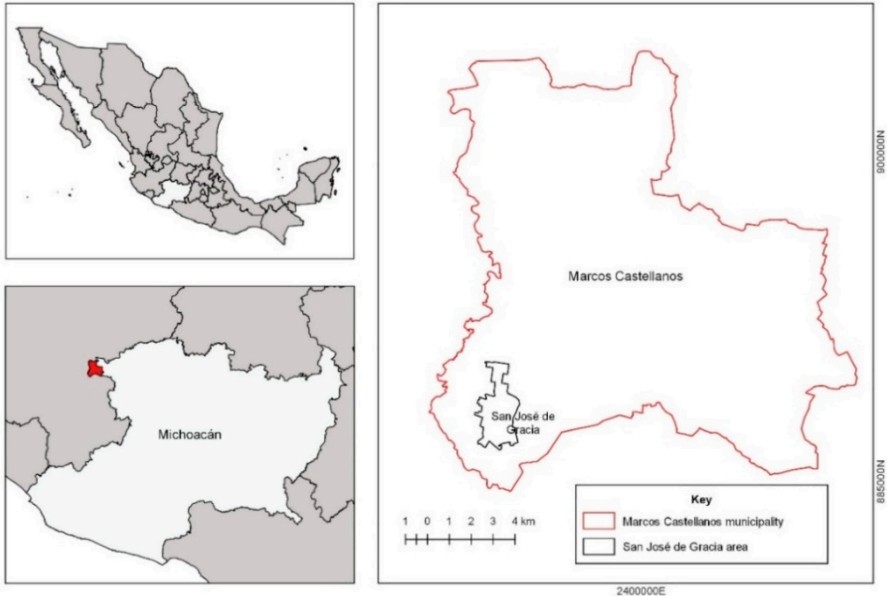

**Figure 1.** Geographic location of San José de Gracia, municipality of Marcos Castellanos, Michoacán, Mexico [49].

### 2.1. Mixed Approach

Both qualitative and quantitative information was collected. The instruments used to collect the information included literature search, review of official databases, participant observation [50] and semi-structured interviews [51] with entrepreneurs. The main items analyzed included types and number of cheese factories by cheese production methods, amount of cheese production, production costs, marketing and type of relationships.

Given the lack of accurate data on the current number of cheese agribusinesses based on the region of study, a census was conducted using the snowball technique [52]. Thirty-seven cheese factories were found, which were categorized according to the general classification of the types of cheeses according to the raw materials used for their elaboration (i.e., natural or imitation cheeses) [24,27]. In the community, it was found according to this classification that there were: (a) 14 natural cheese agribusinesses (37.8%) making cheese from fluid milk, or in combination with powdered milk in the dry season; (b) 11 imitation cheese agribusinesses (29.8%) making cheese by combining both dairy and non-dairy ingredients, partially or completely replacing the fluid milk; and (c) 12 agroindustries producing both natural and imitation cheeses (32.4%), which were named as miscellaneous cheese agribusinesses.

Subsequently, between June and November 2017, 17 semi-structured interviews with participant observation were made to the owners of the cheese factories that agreed to participate in the study. These 17 cheese agribusinesses accounted for 45.9% of the total agroindustry. It was not possible to include the census population of cheese agribusinesses based on the region due to safety and violence issues derived from drug trafficking and organized crime that have struck the country in recent years, with Michoacán being a nerve center of this problem [53,54]. This has restricted this type of research study, since the population are fearful and unwilling to disclose information [55], which was evident in some entrepreneurs who were reluctant to provide information on their cheese agribusiness.

For the interviews and participant observation, a guide and questionnaire were used to gather data about the general characteristics of the agroindustries, as well as the underlying family, inter-company, commercial, technical and financial relationships that each agribusiness had with the different actors involved in the cheese market. The questionnaire was formed with closed questions to obtain quantitative data (production, costs, prices, quantity, frequencies, kind of actors, etc.) and open questions for qualitative information (characteristics of relationships, forms and types of tangible and intangible exchanges, among others). These tools were applied by accompanying the owners and

managers during the production process. With the information gathered on the cheese agribusinesses and their ties, an actor–actor matrix [56] was made in Microsoft Excel. In order to ensure anonymity and easily identify each of the actors, they were assigned a code within the matrix, which in all cases was composed of one or two letters plus a sequence number (Table 1).

**Table 1.** Coding of network actors.

| Actor or Node | Code | Sequence |
|---|---|---|
| Natural cheese agribusiness | N | N1 … Nn |
| Imitation cheese agribusiness | I | I1 … In |
| Miscellaneous cheese agribusiness | M | M1 … Mn |
| Suppliers of raw materials, supplies, machinery, equipment (except milk) | Pr | Pr1 … Prn |
| Milk suppliers | L | L1 … Ln |
| Clients | Cl | Cl1 … Cln |
| Funders | F | F1 … Fn |
| Knowledge and technique | CT | CT1 … CTn |
| Relatives * | Q, IL | Q1 … Qn; IL1 |

\* Relatives were classified into different categories according to the activity carried out: Q refers to a cheese maker located outside the area of study. IL refers to a supplier of dairy products. Cheesemakers, suppliers and family clients located within the area of study were classified into and accounted for in their corresponding category.

### 2.2. Topological and Structural Perspective

In order to analyze the information concerning the social relationships between agribusinesses, a social network analysis (SNA) was used. This is a structuralist tool focusing on the actors and the relationships they have built with one another [52,57] that permits the identification of bonding structures that arise from the various forms of relationship between the different social actors (individuals, institutions, organizations, etc.) [58,59]. These emerging structures are manifested as a social network composed of the various actors and the relationships built by them [60] (p. 25). These relationships have multiple purposes and, given their intermittent and recurrent natures, can be measured and represented in a reticular structure [56,59]. In addition, these relationships produce indicators that explain the social structure and behavior both as a whole and individually [57].

The SNA allowed to analyze the interactions that cheese agribusiness actors of San José de Gracia had with one another in order to understand the functioning of the social structure of this agri-food subsector. To achieve this, centrality measurements (nodal degree and intermediation) and structural or cohesion measurements (density, centralization and reciprocity) were used. The first measurements refer to the position each individual actor occupied in the social network [61] (p. 149), and are a way of measuring the prominence and influence of the actors [62]. The second measurements describe the properties of the structure when considering the network as a whole [61].

Nodal degree: The number of direct links that a particular node has [56,59]. It is calculated through the following equation:

$$C_g(n_i) \; = \; \Sigma^A L(n_i, n_j) / (A - 1) \tag{1}$$

where $C_g(n_i)$ is the number of nodes with which ni is connected, and $(A - 1)$ is the width of the network [63].

Degree of intermediation: The number of times in which an actor appears as a possible connection between a pair of actors that are not directly linked [59]. It is calculated by applying the following equation:

$$C_I(n_i) \; = \; \Sigma g_{jk}(n_i) / g_{jk} \forall j < k \tag{2}$$

where $C_I(n_i)$ is the degree of intermediation; $g_{jk}(n_i)$ is the number of geodesics between nodes $j$ and $k$ that pass-through node $I$; and $g_{jk}$ is the number of geodesics that join the nodes $j$ and $k$ [63].

Density: The percentage of existing ties relative to the total number possible [64]. It is measured by applying the following mathematical expression:

$$\Delta = \frac{L}{g(g-1)} \tag{3}$$

where L is number of existing arcs, and g (g − 1) is possible number of arcs [56].

Centralization: Analyzes the extent to which the ties of a network are concentrated or not in a small group of actors [61] (p. 150). It is obtained by applying the following equation:

$$C_G = \frac{\sum_{i=1}^{N}[C_G(n^*) - C_G(n_i)]}{\max \sum_{i=1}^{N} \cdot [C_G(n^*) - C_G(n_i)]} \tag{4}$$

where $C_G$ is centralization; $C_G(n^*)$ is the maximum value of the network for the $N$ actors in the network; and $C_G(n_i)$ is the value for node $i$ [59].

Reciprocity: Reflects the degree to which the links issued are reciprocated [61] (p. 150). The formula to calculate this indicator is as follows:

$$R = \frac{\#mut}{\#mut + \#asim} R \in [0, 1] \tag{5}$$

where R is the reciprocal links; #mut is the number of mutual connections; and #asim is the number of asymmetric links [63].

The data collected were organized into adjacency matrices and processed with Ucinet 6.645® and Netdraw 2.161 [65] software tools.

## 3. Results and Discussion

The SNA shows that the cheese agribusinesses based in San José de Gracia form a meso-system where different cheese companies converge and coexist, forming a complex social structure composed of many actors (1717). These actors were classified into nine types: natural cheese agribusinesses (14), imitation cheese agribusinesses (11), miscellaneous cheese agribusinesses (12), suppliers of cheese-making supplies, machinery and equipment (35), fluid milk suppliers (244), clients (1367), funders (5), knowledge disseminators/technical assistance providers (18) and relatives engaged in some cheese agribusiness related activity (11) (Figure 2). Regarding the ties between and among the actors, 2190 were found and occurred in various geographic and social scales, ranging from the local and regional to the national.

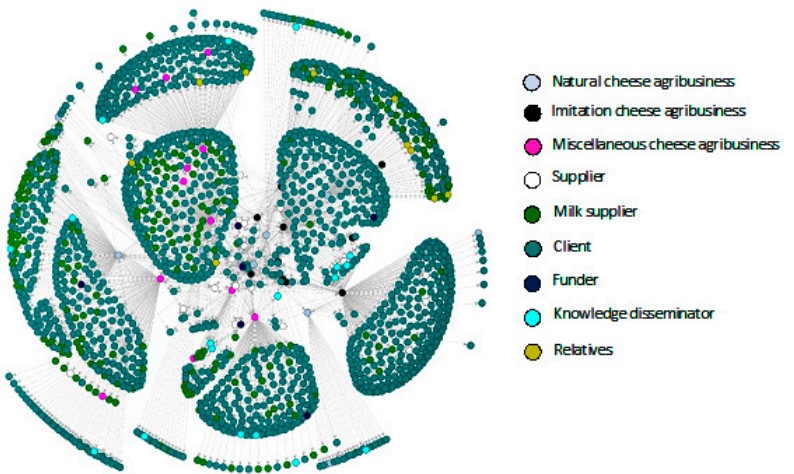

**Figure 2.** Complex social network of cheese agribusinesses based in San José de Gracia.

### 3.1. Components and Scope of the Commercial Network of Cheese Agroindustries

In the complex social network found (Figure 2), it is highlighted that the ties were, for the most part, commercial. Of the total number of actors, 95.9% were clients, suppliers of machinery or equipment and milk suppliers. The greatest number of ties were agribusiness clients (79.6% of the total). The businesses with the largest number of clients were the imitations and miscellaneous enterprises (134.5 and 133.8 on average, respectively, versus 47 average clients for the natural cheese agribusinesses), which is directly related to the market niches, distribution and production volumes of these two types of company: the imitation cheese agribusinesses produced 39,250 kg per day, while the miscellaneous cheese agribusinesses produced 2580 kg.

Clients of the imitation cheese agribusinesses were found to be distributed throughout the national territory; those of the miscellaneous cheese agribusinesses were spread out regionally, and those of the natural cheese agribusinesses were limited to local environments (these latter were the only agribusinesses where the trading part of their production was directly related to the final consumer). The clients had more links with intermediaries, locally known as 'ruteros' (routers), 70 of which were identified in the network. These actors buy cheese of different types (natural and imitation) from the three local cheese agribusiness types and resell them locally and regionally through predetermined distribution routes. These actors were mostly from the community and, in some cases, were relatives of the cheese agribusinesses owners. The main contribution of these ties are in its invigoration of the network, due to these actors' ability to circulate the products, allowing for market diversification.

Likewise, in this social structure, a total of 279 providers were identified, representing 16.2% of the actors in the network, with 244 corresponding to milk suppliers and 35 to suppliers of inputs, machinery and equipment. The number of milk suppliers to the network is strategic in ensuring that the production process does not stop, and in the study it was found that each agribusiness maintained purchase relationships with 14 milk suppliers on average; in other studies, it was found that when suppliers are few, they become dependent, the cost for raw materials rises [28] and quality of milk cannot be demanded [66]. For the agribusinesses studied, the more milk providers had, the safer the supply, the more prices were controlled, and the more they were able to obtain milk of the quality that best suited them, thus achieving better competitiveness [67] and sustainability in the market.

On the other hand, other authors have suggested that sourcing from a high number of providers can make it difficult to build close trust and long-term relationships [68,69]. However, as will be seen, this does not seem to hinder the supply, since there are very close kinship relations between milk suppliers and agroindustry.

In the case of input, machinery and equipment suppliers, an opposite situation occurs. Having fewer suppliers seems to place companies at a greater disadvantage because of such dependence; agroindustries have two providers per company on average. This may explain why a centralization of 5.9% has been found in the network, which indicates that it has concentration symptoms around some individual actors. As a result, most of the agroindustries of the network converge with this type of actor. However, since this type of supplier cannot be controlled, the geographic proximity between companies and their production history has led several supply companies to converge in San José, allowing more than one option for sourcing materials, and freeing companies from having to depend on a single supplier. This has also generated closer and longer-term relationships [68,69].

It may be deduced from these first results that some of the factors that contribute to the coexistence and continued access to the market of the companies are, on one hand, a multifaceted and complex social structure of actors, allowing for market and channel marketing diversification, while on the other hand, a dependable number of clients for each cheese agribusiness. These factors expand their negotiation capacity, reduce their dependence and enable them to get more options for placing their products. Thus, having diversified marketing channels provides an advantageous position of less vulnerability to entrepreneurs, which decreases dependence and competition for customers [70].

From this commercial perspective, the results of the graph (Figure 2) show a density lower than 0.5%, implying a low cohesion between the actors that make up this reticle, as well as limited access

to the information circulating within the network. In this regard, the low probability of interaction between actors is related to three features of the structure: its egocentric morphology, its geographical distribution throughout the national territory and, above all, its commercial nature, which indicates a structure with significant levels of dispersion and low cohesion. Similar results were found in cheese agroindustries in Chiapas, Mexico [29], where researchers showed that as geographical proximity decreases, the density of the network decreases. Similarly, it has been mentioned that, as the social structure grows, the proportion of all possible ties diminishes, since the larger the network, the less likely actors are to know each other [52].

In this sense, a commercial network will always be characterized as a structure of weak ties, where poor values of cohesion will be observed. Therefore, analyzing a social structure only from a commercial and/or technical point of view does not shed light on the internal processes of cohesion/exclusion underlying every social structure and influencing the productivity of the group analyzed.

### 3.2. Analysis of the Cooperative Network of Cheese Agroindustries

In order to address a social structure with greater relational and filial richness of the network of cheese agroindustries in the community, all actors with a nodal degree equal to 1 were discriminated (Figure 3). This procedure was carried out to highlight the linking attributes of the main actors, as well as to gain access to more significant cohesion and socially prominent processes [70].

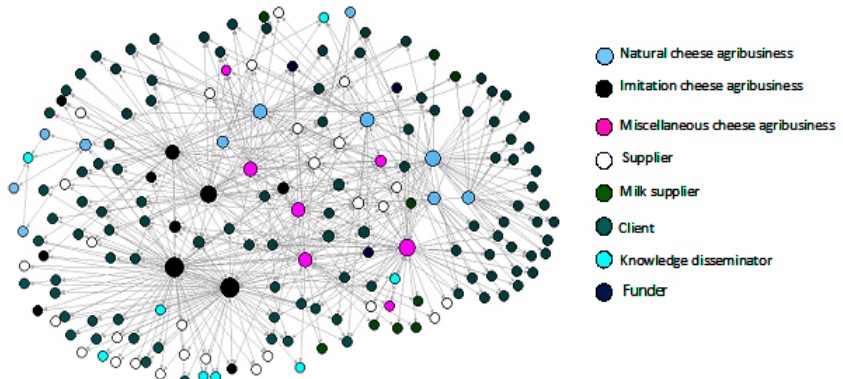

**Figure 3.** Social structure of the cheese agribusinesses based in San José de Gracia (nodal degree >1).

This structural approach revealed a community of 180 actors that were: (a) involved in different links of the cheese agri-food chain, including suppliers, cheese agribusinesses (natural, imitation and miscellaneous), relatives, clients, funders and knowledge disseminators; and (b) tied to one another at different geographic (local and regional) and social scales (Figure 3). This was evidenced by the coexistence and inter-linkage of the various cheese agribusinesses, showing that there were no isolated agribusinesses; on the contrary, they were clearly integrated through multiple ties, creating a 'local neighborhood' of cheese makers in San José de Gracia.

In this regard, the analysis of the structure showed a social density of 1.73%, which, unlike the general structure, indicates greater relational diversification, albeit with a centralization of 43.71%, which is typical of a structure with significant levels of concentration of social prominence around a few actors. These two parameters indicate that some actors controlled a good deal of the relationships and, consequently, had greater negotiating power within the network. This can be observed in Figures 3 and 4, where agglomeration is greater around some actors, mainly imitation cheese agribusinesses.

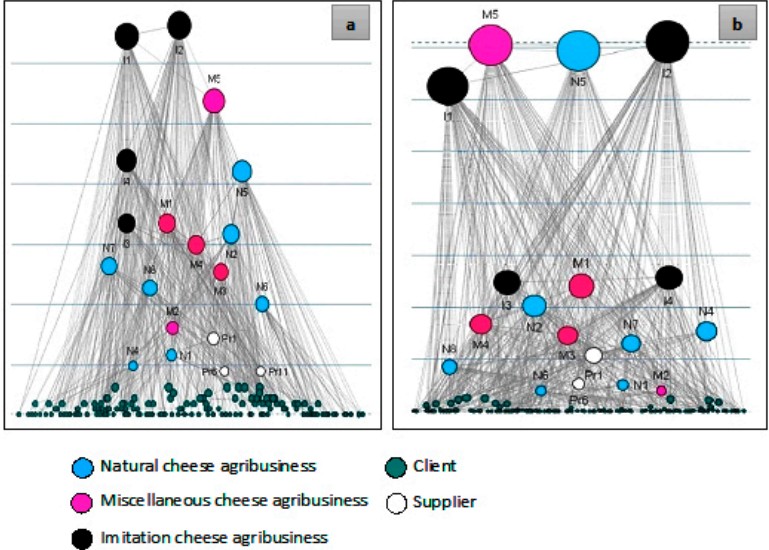

**Figure 4.** Nodal degree (**a**) and intermediation (**b**) in the social structure of the cheese agribusinesses of San José de Gracia.

### 3.2.1. Social Prominence in the Cooperation–Competence Network of the Cheese Agroindustry

According to the calculation of the nodal degree (Figure 4a), the actors with the greatest strategic access to financing, knowledge and social recognition flowing within the structure examined were, in order of importance, imitation cheese agribusinesses (I1, I2, I4, I3), followed by actors from the miscellaneous cheese agribusiness group (M5, M1, M4, M3) and the natural cheese agribusinesses (N5, N2, N7, N8, N6). Particularly, companies I1 and I2 represented the most significant nodal degrees (6.4% and 6.6%, respectively), due partially to the fact that they were the largest and most technified, with their joint production amounting to 38,300 kg per day, 86.46% of the total produced by the 17 companies included in the study. In addition, due to their technical and productive conditions, these agribusinesses had the lowest production costs (1.06 and 0.23 euros per kg, respectively) and the highest unit profits, more than 0.16 euros per kg sold. Companies M5 (5.4%), I4 (4.4%) and N5 (4.2%) also stand out, since although they did not have as many direct links as companies I2 and I1, they had a considerable number of relationships that also put them in situations of greater advantage compared to agribusinesses that lack such a high number of relationships.

The cheese agribusinesses I2 and I1 had the largest number of clients (244 and 196, respectively), the majority of technical assistance (food engineer and private and institutional advice in productive and administrative processes) and the greatest number of non-dairy suppliers (18 and 16, respectively). All of this allowed them to negotiate better conditions for purchase prices, terms of payment and acquisition and placing of products. These types of factors, according to Porter [39], enhance competitiveness.

Hanneman and Riddle [52] have argued that actors that maintain the greatest number of ties with other actors, such as I2, I1, M5, I4 and N5, have the possibility of placing themselves in more favorable positions by being able to rely on a wide range of options. Due to this, these companies have alternative ways to meet their needs and be less dependent; therefore, they can access and leverage more network resources [64].

The intermediation analysis of the cheese agribusinesses shows that the mediating actors that stand out for their ability to link and influence various actors within the network and, therefore, the flow of social information, are imitation, miscellaneous and natural cheese agribusinesses, where I2 (19.1%), N5 (10.92%), M5 (9.7%) and I1 (7.6%) stand out (Figure 4b). In this case, a different social grid is observed, although similar to Figure 4a, where actors I2, M5, N5 and I1 were also actors with a high nodal degree. In this regard, it is suggested that in many cases there are actors who can have a significant nodal degree and, therefore, also stand out for their degree of intermediation [71] (p.19).

This allows them to position themselves as actors with great power, as they not only have a high number of ties, but are also linked with actors with few or no alternatives to connect directly with other actors in the network, so they can decide whether to connect with them or not. Also, they have information that flows through unconnected actors or groups of actors. In this sense, the active participation of actors in social networks is a critical factor to improving their status within society, since it allows them to be positively perceived by it [72].

These same actors are recurrently found in the various measurements obtained, which indicates that only a few actors control the resources that flow within the network, such as a greater capacity for negotiation and influence, a significant social interrelation and access to varied information, all of which grant them advantages to compete for knowledge and power [44,71]. However, as Long [73] (pp. 341–342) suggests, the actors with power do not have complete control over the scenario, and their power is forged by ties with other actors, with links being the results of family and other relationships (friends, neighbors, colleagues) and expertise they have conceived and accumulated through the years [74].

### 3.2.2. The Family and Friendship Network: Trust, Reciprocity and History among Cheese Agroindustries

Several studies have reported that formal institutions (government and universities) are key actors for the generation of knowledge and innovation in companies [75–78]. For the enterprises analyzed in this study, while institutional support has not been absent from their development, the structural configuration of this network has generated a system of diffusion and adoption of innovations through informal relationships (without a legal contract or agreement), mainly by the flow of information, transfer of technology, the market and technology to improve production processes. This has given the network self-management capabilities to access knowledge and information on the market and on the operation of commercial networks, which implies that these companies collectively shape their knowledge through informal exchanges of knowledge and experiences, and provides evidence of a network nurtured by bonds of cooperation and trust. In spite of this, the study on how knowledge and innovation networks are produced in the dairy and cheese agroindustry is a pending issue to be investigated, since there are few studies, and these are focused on primary production [79–84].

Thus, the knowledge and power of actors are not only built on their commercial, financial and technical relationships, but are also based on kinship and friendship interrelationships with other companies—that is, on all social situations [73]. In this regard, some companies' owners have several relatives who are also engaged in the production of cheese (47.1%), as was the case for the owners of I2, M1, M4, M5, M6, N1 and N4. In general, the cheese makers who were part of the same family reported exchanging information on prices (e.g., fluid milk and other supplies such as rennet, salt, milk powder and vegetable fat), productive performance and some failures in production processes. Additionally, they often loan one another supplies, equipment and workwear, including skimmers, curd knives, cheese-making tubs and even finished products. These are bonds based on trust, cooperation and reciprocity between and among the parties.

Some of the relatives of the entrepreneurs were milk producers, suppliers and cheese retailers who supported the cheese agribusinesses as suppliers or clients. For example, the owners of companies I1 and I2 were relatives in the first degree of Pr1's owner, which was the main supplier of supplies, machinery and equipment in the area of study, and was also the preferred supplier of entrepreneurs for purchasing everything they require for manufacturing their products; in fact, 16 of the 17 cheese factories studied considered it as a supply option. This infuses a certain degree of stability to enterprises by offering a fixed supply of inputs, milk as an essential raw material and an entry route into the market for their products. In addition, these actors exchanged information with cheese makers on market prices and on competition.

Table 2 and Figure 5 show that the companies with the most relatives engaged in some activity related to the agribusiness were the miscellaneous ones (6.8 relatives on average), engaged mainly

in making or reselling cheese. In addition, it was found that 76.6% of entrepreneurs had family members involved in some activity of the cheese agri-food chain with whom they maintained daily relationships. This demonstrates that both family character and the historical tradition [46,48] are linked to the permanence of cheese activity in the municipality, since, as stated by the entrepreneurs, this is what they have done for a good part of their lives and what they know how to do. On average, the entrepreneurs have been making cheese for 26.6 years, and 70.6% of them said they had gained the knowledge required to make cheese through family ties.

**Table 2.** Family and intercompany relations among cheese agribusinesses based in San José de Gracia.

| Cheese Agribusiness | No. of Companies | Average Age (years) | Monthly Average Production (kg) | Liters Milk Processed per Day (average) | Relatives Engaged in Cheese Agribusiness (average) | Ties with Unrelated Cheese Agribusiness (average) |
|---|---|---|---|---|---|---|
| Natural | 8 | 19.5 | 9416.9 | 2600 | 1.4 | 2.8 |
| Imitation | 4 | 28.3 | 299,281.3 | 2237 | 2.3 | 5.0 |
| Miscellaneous | 5 | 25.6 | 15,738.0 | 3500 | 6.8 | 4.2 |
| Average | - | 23.4 | 79,479.4 | 2779 | 3.2 | 3.7 |

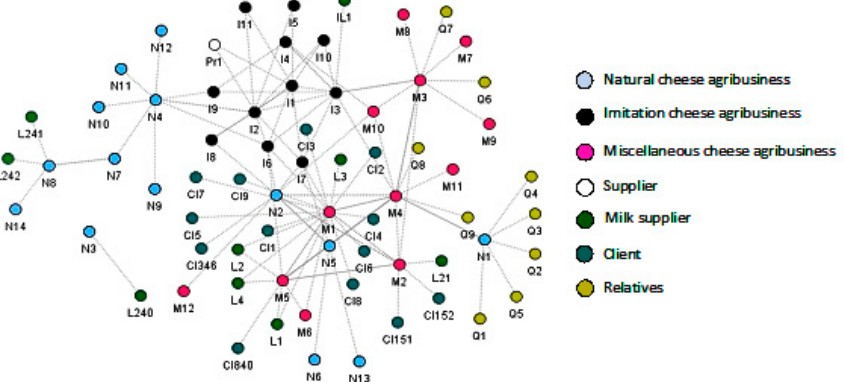

**Figure 5.** Network of family ties and friendships in cheese agribusinesses based in San José de Gracia.

Figure 5 shows the network of kinship ties. Note that companies M5, I1 and I2 have a significant amount of these types of ties, and their high degree of intermediation results, in part, from these types of ties that allow them to better access the information flowing through the network. At the same time, they build emotional and supportive relationships that create greater cohesion between them, which is evident by a higher density (2.6%). Through these ties, companies gain a certain amount of power; however, while these actors seem to have control over the scenario, such control is shaped by those who do not seem to be in control and act as subaltern actors [73] (p. 342).

Another important aspect to be noted is the ties that cheese agribusinesses maintain with unrelated counterparts (Table 2 and Figure 5), as 88.2% of entrepreneurs had relationships with other companies. The results suggest that these ties may have been related to the age of companies (Table 2): the older the cheese agribusinesses, the more ties they have with other cheese factories, and therefore a greater informal social integration into the network will develop over time [57]. Also, interrelationships were found between different types of companies (e.g., companies N2 and N5 maintained relationships with the three types of companies: natural, imitation and miscellaneous).

The relationships the cheese agribusinesses form with one another are based on an exchange of information, mainly on the prices of fluid and powdered milk, but also on raw materials such as rennet, salt, vegetable fat and starch. Some of them usually exchanged information on their yields, production process issues, the situation of the cheese agroindustry, consultants for the production process and supplier data. Consequently, those companies that related both to cheese agribusiness of their kind and of a different kind had a more comprehensive information overview. Something similar

was reported of the agroindustry of Chiapas [29], where it was identified that companies that were geographically closer had a greater exchange of information; however, despite proximity, this does not always happen. In a study carried out in Aculco, State of Mexico [44], an asymmetry was reported in the networks of cheese makers and family relations, whereby intense competition for hoarding suppliers and customers prevented agroindustries from sharing information, and therefore, there was not enough cooperation between them. As such, proximity cannot be the only feature that allows cooperation, and should be accompanied by access to different market segments and the maintenance of trusting relationships based on kinship, friendship and reciprocity, as observed in San José de Gracia.

The ties between entrepreneurs and their families as well as other agribusinesses reveal the existence of reciprocal ties. Figure 6 shows these types of ties maintained among actors in the network (highlighted in red), where 136 reciprocal arcs were found. Among these ties, are remarkable the tangible exchanges between companies with no family bonds: reciprocate loan of raw materials, machinery and equipment and even final products, as well as, sometimes, the sale of these things. These exchanges foster future interactions and build and maintain lasting relationships [85].

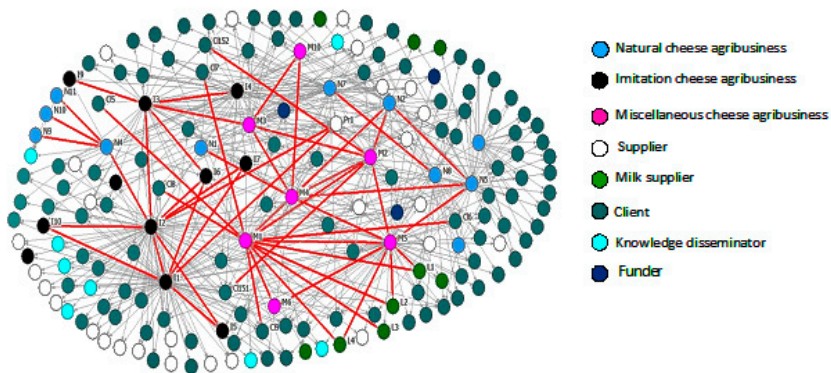

**Figure 6.** Reciprocal ties in the social structure of cheese agribusinesses based in San José de Gracia.

According to Molina and Ayalo [85], reciprocity-based exchanges tend to build long-term relationships. In this respect, it was found that, in general, the age of agribusinesses may be directly related to a greater number of ties or bonds with other companies, whether owned by relatives or non-relatives (Table 2). Imitation cheese agribusinesses have been in the market for an average of 28.3 years and maintain, on average, five ties with companies owned by non-relatives. It has been claimed that cooperation and trust between companies are largely the result of the process of reciprocal relationships that companies have built over time, and that derive from the natural socialization that occurs due to the proximity or concentration of the actors in a region [34,37,42,86]. This is evident in the sample studied, since 58% of the current owners inherited the ownership of their cheese agribusiness; that is, they belong to the second generation, and indicate that their predecessors already had ties with one another. In San José de Gracia, where the community is geographically and historically demarcated [46], 'face to face' relationships are constant, which cause reciprocity and trust to emerge, thus creating a common culture [76]. This cohesive social fabric helps companies to endure [87,88].

According to Ayakwah et al. [37], information, knowledge and collaboration circulate through informal relationships and horizontal community ties based on friendship, kinship, reciprocity and local and cultural history. This helps cheese agribusinesses to reduce uncertainty and cope with and prevail in the face of high competitiveness in the cheese market, without neglecting their business objectives (i.e., to maximize profits and minimize losses), which allows the permanence, coexistence and development of companies [36].

Hence, these relationships do not occur because of, nor are they maintained only through, sales and power relations, but also through the reciprocal ties of trust and support that have been formed among actors over the years. This is in contrast with several authors who describe the communities of cheese producers as distrustful, disorganized, isolated and lagging in innovation, and who consider

commercial organization through formal contracts (associations, cooperatives, collective marks) as the only way for collaboration to obtain actual benefits in order to remain in the market [4,28,31].

Thanks to the combination of cooperation and competition between actors that underlies the social and commercial structure of the cheese agribusiness chain, the manufacturing sector presents companies with greater solidity and sustainability [87] (p.11). These forms of social organization that allow the creation of new mechanisms of collaboration and solidarity facilitate the formation of a social fabric that keeps companies functioning in the market. These features may explain why Michoacán in Mexico is one of the states that have the largest number of manufacturing companies with more than 15 years of life [6] (i.e., has more social and cohesive companies).

## 4. Conclusions

The cheese agribusinesses based in San José de Gracia form a meso-system, where three types of cheese factories converge and coexist (natural, imitation and miscellaneous) to make up a complex social structure composed of 1717 actors with a predominance of commercial actors and relationships generating commercial exchanges locally, regionally and nationally. This structure is marked by weak ties between the actors and a lack of connections between many of the network's components (<0.5%). In this network, it was observed that some of the factors determining the coexistence of the companies in the area, as well as their continuity in the markets, include their differentiated offerings, their distribution channels, the segmentation of markets and their wide extensive client portfolios, all of which suppress competition among them.

It was evident that among the diversity of cheese agroindustries there is a deeper social structure based on long-term relationships of kinship, friendship and trust that translates into reciprocal exchanges of goods and services providing greater cohesion to face the uncertainties of the market. These exchanges occur among a certain number of actors and relationships operating at different organizational and commercial levels to create synergy and interdependence among them, in order to exchange tangible and intangible goods and services despite asymmetries and differences.

Thus, this structure is marked by both commercial (suppliers and clients) and cooperative relationships between and among the agribusinesses and the family members engaged in some link of this chain of value. This means that this social structure behaves like a 'local neighborhood', where everyone knows each other and coexists, competes and shares with one another. Therefore, social prominence is managed, allowing companies to be sustainable in a highly competitive market.

The study showed that the social structure of cheese agroindustries in the community, despite their asymmetry, managed to be economically sustainable in a globalized and highly competitive environment through cooperation–competition relations without a need for formal agreements. This study provides important lessons for institutions that promote competitiveness and local development, because the results show that in order to make companies competitive and sustainable in the market—besides stimulating competitive advantages of geographical proximity—it is essential to understand social structures and promote relationships of trust and friendship established over time is essential, as in the case of San José de Gracia. For this reason, if commercial collective action is the only way of collaboration to obtain real benefits for rural agroindustries, it should advocate identifying mechanisms that foster trust, friendship and reciprocity in territories.

Likewise, this study could contribute and be a reference for future studies that evaluate the economic and social sustainability of business conglomerates. Based on empirical evidence, it was possible to identify that the affective and reciprocal aspects of social and business relationships, as well as the geographical proximity could be considered as attributes to provide sustainability to companies in the face of the effects of economic globalization.

**Author Contributions:** Writing—original draft, M.C.R.-R. and L.M.C.P.; conceptualization, V.E.E.O. and R.S.-R.; investigation, M.C.R.-R.; data curation, M.C.R.-R. and J.F.N.E.; formal analysis, M.C.R.-R., V.E.E.O. and R.A.J.-J.; writing—review and editing, L.M.C.P., R.A.J.-J., R.S.-R., V.E.E.O. and J.F.N.E. All authors therefore approved the final manuscript.

**Funding:** This research was funded by the Program to Support Research Projects and Technological Innovation (PAPIIT) grant number IN309317 of the National Autonomous University of México.

**Acknowledgments:** The authors would like to acknowledge the entrepreneurs who used their precious time to participate and make this research study possible, and we want to express our appreciation to the National Council on Science and Technology (Conacyt) for the doctoral grant awarded to the first author.

**Conflicts of Interest:** The authors declare no conflict of interest.

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
