# Peer review of "The Social Fabric of Cheese Agroindustry: Cooperation and Competition Aspects"

_sustainability, doi:10.3390/su11102921_

Reviewer 1 Report

Overall, the paper provides an interesting approach to the Social Network Analysis by employing it to identify and assess the social framework of the cheese agroindustry of a Mexican region. The research is within the scope of the journal, it offers meaningful results and provides some potentially new and important information. Nevertheless, I think that the paper is quite extensive, and it could be shortened to a certain extent, especially when it comes to the concluding section; The conclusion is not a discussion. The author(s) should try to summarise by limiting the results to only one paragraph and focus more on the policy implications and the paper’s novelty versus current knowledge. Few points for minor amendments follow:

Introduction: Clarify the novelty of the study versus current knowledge

Page 5, line 173: it is better to write “census population” instead of “the entire universe”.

Page 5, line 178: provide more details about the interviews, especially the questionnaire and what type of questions were included.

References: I believe that an update to the references list could be useful as the majority of the references are in Spanish and is difficult to follow parts of the literature in the article.

Author Response

Dear reviewier:

We have uploaded to answers in a Word file.

We appreciate your revisions.

Kind regards.

Reviewer 2 Report

This study looked at the social structure aspect of a community by navigating the community's cheese industry - which according to the article, is considered a pillar for the community and continues to be its major income source. The research is somewhat interesting with minor grammatical errors, however, what I didn't find is the actual value of the research. Why is this research significant to the community, to the cheese industry, to national interest, perhaps? Does it have a socio-economic importance to a wider scheme of things? Why should other researchers, or stock-holders, for that matter care? What is this study contribution to social science or to your field? There are several instances where the authors could have defined/stated this important information, either in the abstract or conclusion (see details below).

Abstract:

Line 20: "...structure of the cheese factories (towards?).." what's your ultimate aim?

Lines 21-22: "Participant observation and semi-structure interviews - are these surveys? Is yes, say so.

Line 25: "natural, imitation, and mixed" - I've never heard these terms in the cheese industry.

Lines 26-30: What does this imply?

Line 27 and elsewhere: "actors" - not sure if this is the correct term. Is this a role?

Introduction: - it is way too long; what is the focus of your study? I think that this introduction contains trivial information that is not directly related to the study.

Lines 37-41: This is very long sentence and it's confusing. Either state the three organization trends first in one sentence and then define each trend in succeeding sentences or state and define each trend per sentence. As currently stated, I don't understand which is the actual trend and which one is the definition of a specific trend.

Lines 46-49: This is not a complete sentence.

Lines 61-62: "globalization of global agrifood systems" - sounds redundant.

Line 67: "This caused great.." - What does "This" refer to? Specify.

Line 70: "An effect of the above.." What does "above" refer to? Specify.

Line 70-76: So what? How does this paragraph help in the overall understanding of your study?

Lines 130-133: For what? So what? Towards what? - ultimate aim?

Material & Methods:

Line 154 and Line 157: Either use "combined" or "mixed". Choose one and be consistent.

Results & Discussion

Line 240 (and elsewhere L245, 250, 279, etc): "actors" or "nodes" - again, choose one and be consistent.

Line 244: What are "ties"? Define.

The rest of the discussion is not exciting to read. I would recommend to just highlight the actual results and then discuss their importance and build on them towards the whole/ultimate aim of the research.

Conclusions:

Line 477: "In the network with a greater degree than one.." - what does this mean?

Lines 477-480 vs Lines 480-483: What is your point of connecting these contrasting ideas? I don't understand the juxtaposition of these 2 scenarios in this paragraph.

Lines 491-499: Now that you know this, then what? (See above general comment about this article).

Author Response

Dear Reviewer:

We have upload the answers in a Word file.

We appreciate your reviews.

Kind regards.

Reviewer 3 Report

Authors in the present study have analyzed the social structure of cheese factories located on an agribusiness territory of Mexico (San José de Gracia) using Social Network Analysis. It is a well-constructed manuscript with very interesting outputs presenting the family, intercompany, commercial and technical ties which are built among them. Only some minor comments are to be checked prior publication.

1.     Authors are advised to modify their title as “The social fabric of cheese agroindustry: cooperation and competition aspects”.

2.     Line 34: Why is “imitation cheese” considered to be as important to be a keyword? In general authors are advised to revise their keywords as more “to the point ones” could be presented.

3.     Lines 30-33: Authors are advised to provide a comment regarding other similar economies.

4.     Lines 70-76: Is this a model of a dairy market, composed of three types of companies meet in other countries as well? Add some more information regarding this matter. Has Social Network Analysis been applied for dairy industries before? Please add this in the discussion section as well.

5.      Lines 79-80: Is whey a product that can be used also? If so, please add it with the appropriate reference.

6.     Line 87: what does this mean “(Anonymous, cited by Bachmann)”?

7.     Line 133: References can be avoided since the target of the study is presented.

8.     Lines 142-150: This part can be moved to the introduction section

9.     Lines 162-169: authors could provide a list (Table) of the cheese industries they studied provided by their capacity. Why did the authors choose in total 37 cheese industries? Was that the total of industries of this area?

10.  Lines 187-19: Please minimize this part. This discussion could be added in the discussion section.

11.  Line 248: Please use bigger letters in Fig. 2 as it can not be read as it is.

12.  Line 267: a significant number that does what? Please specify.

13.  Lines 331-332: Please check grammar. This sentence is not clearly presented.

14.  Table 2. Please replace # with No.  

15.  Suggestion: The conclusion section can be minimized.

Author Response

Dear Reviewer:

We have upload the answers in a Word file.

We appreciate your reviews.

Kind regards.

Round  2

Reviewer 2 Report

I appreciate the authors' efforts in revising the manuscript. A few more comments as follows:

Line 20: The aim of this study or research... not document.

Line 26: "..natural, imitation, and mixed" - I appreciated the authors explanation, however, as someone who knows the cheese industry, a technical paper such as this one should be careful in using terms that are, not technical or at least used by locals.   

I understand your explanation but I don't think that is objective since you "chose to make your own classification" - this is just adding to the confusion of the already complex world of the dairy/cheese industry. I would maybe use the terms that locals are using and then incorporate them to your explanation.

Line 25 delete San Jose de Gracia and Line 32: delete "..of San Jose Gracia cheese agribusiness.." - and replace with "of this community" - to make it more succinct. We already know which community you're talking about anyway since it was previously mentioned in line 24.

Introduction

Lines 40 to 44 - still is a long sentence. What is your point? Is this good? Is this bad? Is it complicated? So state that. You can say like: "The establishment of the neo-liberal model brought a complex/complicated/sustainable/unsustainable/(or what have you) socio-economic structure. These societies are defined by (1) an extreme deregulation of economic acitivities…; (2)… and (3)…"

Lines 49-51: Still not a complete sentence - it cannot stand on its own and therefore does not make sense. I understand that it is an example that you want to make out of the previous sentence but the way it's written is not quite right. Revise.

Line 67;.."The above.." you got to be more specific and make this manuscript more reader-friendly. You know exactly what you're talking about but readers don't necessarily and automatically know which one/s you're referring to. 

Line 78:.. "the most advanced"...this could be debatable and I wouldn't necessarily call it the "most" advanced but relatively "more advanced technology". In the grand scheme of things, technology always evolves and a more develop technology could be generated/produced/developed.

Lines 147 - 148: "As observed in another type of agribusiness and production chains mainly in Aftica [37,40,41,42,43]. This is not a complete sentence.

Results and Discussion:

I think this whole section could benefit on using subheading/subdivision so it does not look like a one lackluster, continuous discussion. It'll be much easier to follow what result is being discussed at the moment if the individual points/results are divided into mini-subsections.

Conclusions;

Line 536: "In this network was observed..." check the grammar.

Lines 539 to 543: Too long of a sentence, delete "Additionally, with the discarding of actors with nodal level equal to 1..." and just start the sentence with "It was evident that..."

Line 541: "based on ong-term.." check the spelling

Line 549:  "All of this causes.." check the grammar.

Line 552: "The study shows from a different approach and little explore.." This does not make sense.

Line 555: "...without the need for contracts formal" - what is contracts formal? Also, this sentence is missing a period.

Lines 556-562: This is good. This has to be moved up/incorporated in the abstract and deconstructed to be part of the objective in the last part of the Introduction as it gives the readers the "why" and "importance" of this research. I had to scroll down to the very bottom of this manuscript to actually understand the importance of the study (at least to the researchers) which means that if this is the goal of the research, then this has to highlighted and stated early in the manuscript.

Lines 563: Although I appreciate this sentence, as it states what application/contribution this study offers, it is way too long and general. Break it into 2-3 sentences and make it a little bit more specific.

Author Response

Dear reviewer:
Thank you very much for your comments and observations.
We attach a file where we answer each of the points you indicated.

Kind regards
